# Nutrimedia: A novel web-based resource for the general public that evaluates the veracity of nutrition claims using the GRADE approach

**Montserrat Rabassa[1], Pablo Alonso-Coello[1,2]\*, Gonzalo Casino[1,3]\***

**1** Iberoamerican Cochrane Centre, Biomedical Research Institute Sant Pau (IIB Sant Pau), Barcelona, Spain, **2** CIBER de Epidemiología y Salud Pública (CIBERESP), Barcelona. Spain, **3** Departament of Communication, Pompeu Fabra University, Barcelona, Spain

\* palonso@santpau.cat (PAC); gonzalo.casino@upf.edu (GC)

## Abstract

The objective of Nutrimedia is to evaluate, based on the scientific evidence, the veracity of nutrition claims disseminated to the public by the media. In this article, we describe the methodology, characteristics and contents of this web-based resource, as well as its web traffic and media impact since it was launched. Nutrimedia uses a systematic process to evaluate common beliefs, claims from newspapers and advertising identified and selected by its research team, as well as questions from the public. After formulating a structured question for each claim, we conduct a pragmatic search, prioritizing guidelines and/or systematic reviews. We evaluate the certainty of the evidence using the Grading of Recommendations, Assessment, Development and Evaluation (GRADE) approach, and classify the veracity of each claim into seven categories (true, probably true, possibly true, possibly false, probably false, false, and uncertain). For each evaluation, we develop a scientific report, a plain language summary, a summary of findings table, and, in some cases, a video. From November 2017 to May 2019, we published 30 evaluations (21 were related to foods, six to diets, and three to supplements), most of which were triggered by questions from the public (40%; 12/30). Overall, nearly half of the claims were classified as uncertain (47%; 14/30). Nutrimedia received 47,265 visitors, with a total of 181,360 pages viewed. The project and its results were reported in 84 written media and 386 websites from Spain and 14 other countries, mostly from Latin America. To our knowledge, Nutrimedia is the first web-based resource for the public that evaluates the certainty of evidence and the veracity of nutrition claims using the GRADE approach. The scientific rigor combined with the use of friendly presentation formats are distinctive features of this resource, developed to help the public to make informed choices about nutrition.

## Introduction

Most common chronic diseases such as cardiovascular diseases, cancers and diabetes are leading causes of death, accounting for 71% of all deaths and 43% of the global burden [1]. Modifiable behavioural risk factors such as poor eating or food habits increase the risk of chronic

**Data Availability Statement:** All the results of the evaluations are available in the corresponding technical reports freely accessible on the Nutrimedia website (https://www.upf.edu/web/

nutrimedia); all referenced papers are available in the usual databases (MedLine, Cochrane Database of Systematic reviews, etc.). Data of the media impact from Factiva can be reproduced by a search with the term "Nutrimedia". Data on web traffic from Google Analytics is not legally or ethically restricted and it is not necessary to replicate the results –it changes constantly with time.

**Funding:** Nutrimedia has been partially supported by the Spanish Foundation for Science and Technology (FECYT) grants from the Spanish Ministry of Science, Innovation and Universities (FCT-16-11294 and FCT-17-12460). The funders had no role in the study design, data collection and analysis, decision to publish, or preparation of the manuscript.

**Competing interests:** The authors have declared that no competing interests exist.

**Abbreviations:** CPG, Clinical Practice Guideline; GRADE, Grading of Recommendations, Assessment, Development and Evaluation; PICO, Participant, Intervention, Comparison and Outcome (s); RCT, Randomized controlled trial; SoF, Summary of Findings; SR, Systematic Review.

diseases [2]. Raising awareness of scientific knowledge about food and nutrition among the public can help improving health overall, and prevent chronic diseases [3].

The dissemination of nutrition information in the media is a potential mean for promoting knowledge about appropriate food choices [4–5]. Media (including radio, television, newspapers, internet and social media) today include a wealth of information about food and nutrition. For example, a recent Google search showed over half a million results related to the terms "nutrition advice". Media has been shown to have a potential impact on knowledge and awareness of the public of health issues related to the field of nutrition [6–7]. However, this information is often misleading and contradictory [8–10]. According to recent research, the public is regularly exposed to mediocre or poor quality information about nutrition on websites [8] as well as in newspapers [9]. Approximately two-thirds of newspapers related to dietary advice are based on low-quality scientific evidence [10].

Online resources or websites (e.g., Google searches, YouTube) were the most popular source of nutrition information, between 2003 and 2018, among adults [11–12]. However, to the best of our knowledge, there are no online resources that formally evaluate and communicate the veracity of contemporary claims about nutrition based on scientific evidence. Therefore, we have developed a nutrition web-based information resource for the general public named Nutrimedia that provides rigorous evaluations of claims about nutrition. In this article, we describe how we have developed this resource and present an overview of its characteristics, contents and media impact. The research protocol is available in Spanish from the authors upon request.

## Methods

### Identification and selection of nutrition claims

To identify claims, we developed a preliminary list of claims based on news from the media, Google searches of current topics in nutrition and our knowledge and expertise. This preliminary list has been updated over time.

We classified the topics identified in the following four types of claims:

- Common beliefs related to the health effects of certain diets, foods or nutrients.

- Claims in pieces of news published in newspapers that report about nutrition or nutrition research (newspaper claims).

- Advertising claims in traditional or online media (newspapers, magazines, radio, television, websites, blogs and social networks) about the effect of a particular food (or an ingredient) on consumer health or performance.

- Claims based on questions from the public collected via an online survey posted on Nutrimedia website (e.g., the question "Is meat carcinogenic?" becomes the claim "meat is carcinogenic").

Common beliefs and claims from newspapers or advertising were selected from the preliminary list based on two criteria: 1) achieving the highest interest score (we scored on a 5-point Likert scale the interest of each claim), and 2) ensuring a balanced selection of different types of claims and foods.

To collect questions from the public, we published an online survey from November 20, 2017, until May 4, 2018. The survey included 10 closed-ended questions selected from the preliminary list of claims (to be rated by users on a 5-point Likert scale from 1–5; definitely not to definitely yes interested), one open-ended question and space for comments and suggestions. The questions from the public that were evaluated were selected from those most highly rated

among users in the 10 closed-ended questions and those posed in the open-ended question (the selection was made considering interest and feasibility).

### Scientific evaluation of nutrition claims

For each claim selected, we developed a scientific report using a systematic and explicit process shown in Fig 1 that includes:

**1. Formulation of structured clinical question.** For each nutrition claim, we formulated a structured clinical question in a PICO (participant, intervention, comparison and outcome (s)) format. We included a maximum of four key public-important outcomes (e.g., death, cancer or cardiovascular events).

**2. Identification and selection of the evidence.** To identify and select the best available scientific evidence, we conducted a search prioritizing clinical practice guidelines (CPG) and systematic reviews (SR). We searched on electronic databases (e.g., we searched in MEDLINE and Cochrane Database of Systematic Reviews for SR) (S1 Table), combining at least one MeSH term and free-text terms, with title and/or abstract restrictions, for each PICO component. We also searched grey literature (e.g. Google Scholar), governmental and institutional sources (e.g. World Health Organization, U.S. Department of Agriculture, Spanish Agency for Food Safety and Nutrition) and scientific societies websites (e.g., Sociedad Española de Nutrición Comunitaria).

We included CPGs and/or SRs that used systematic methods to search and identify the evidence in two or more databases (e.g., MEDLINE, EMBASE) and that evaluated the risk of bias

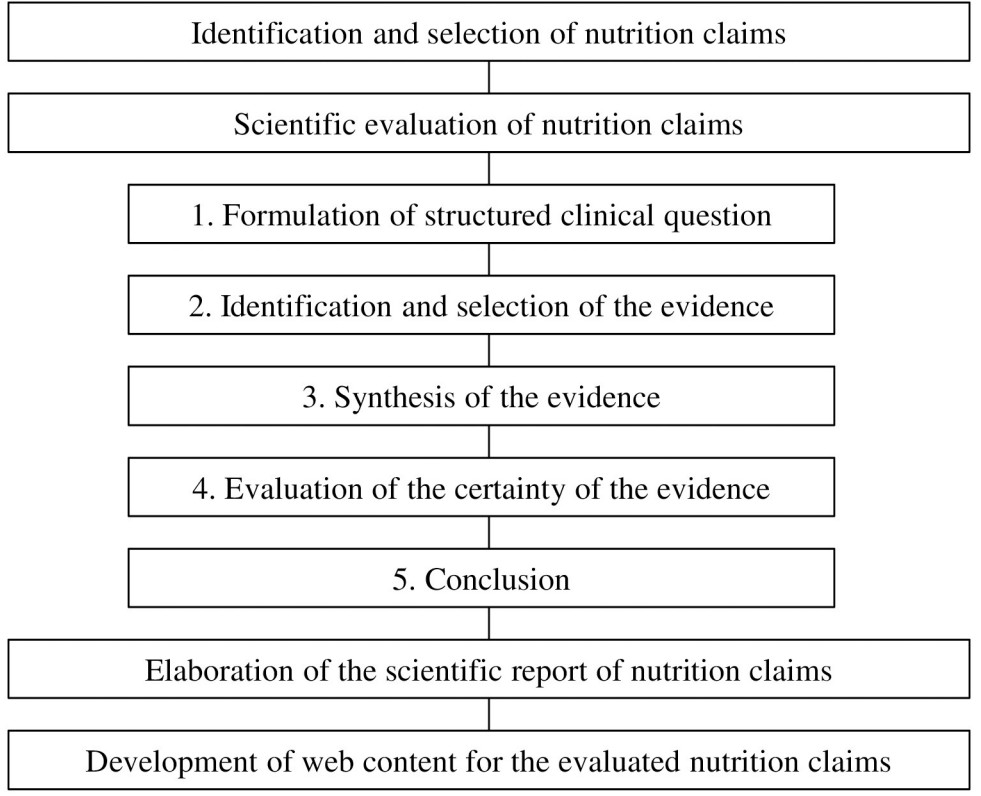

**Fig 1. Nutrimedia evaluation process.**

of research evidence [13]. When we obtained more than one CPG or SR of similar quality we prioritized the most recent. If not available, we included primary studies prioritising randomized clinical trials.

**3. Synthesis of the evidence.** We summarized the following information of each of the documents selected: 1) objectives, 2) methods, 3) main results, and 4) conclusion of the authors (if applicable) (S2 Table).

**4. Evaluation of the certainty of the evidence.** We assessed the certainty of the body of evidence (CPG or SR or primary studies) about the effects of nutrition interventions using the GRADE approach [14]. GRADE classifies the certainty of evidence as high, moderate, low, or very low for each outcome of interest within the same clinical question. GRADE categorizes randomized controlled trials (RCTs) as high certainty, whereas observational studies start as low certainty. An available body of RCTs can be downgraded on the basis of five factors: study limitations (risk of bias), indirectness, imprecision, inconsistency of results, and publication bias. In the case of observational studies, one can consider rating up on the basis of three factors: large magnitude of an effect, dose-response gradient, and plausible residual confounding (Table 1).

For each evaluation, we generated a Summary of Findings (SoF) table using GRADEPro [15]. SoF tables provide a structured outline of the number of studies and number of participants included for each outcome of interest, certainty (or confidence) of the evidence, and the results both in relative and absolute terms (S3 Table). If the certainty of evidence differed across critical outcomes, the overall certainty of evidence is the lowest certainty of any of the public-important outcomes [16].

**5. Conclusion.** We classified the veracity of each claim according to the certainty of the evidence into seven categories: true, probably true, possibly true, possibly false, probably false, false, and uncertain. For example, we assigned true or false, probably true or false, possibly true or false and uncertain conclusion when the overall certainty of the evidence related to a claim was high, moderate, low and very low (or no studies), respectively (Table 2).

## Development of web content for the evaluated nutrition claims

For each claim, we developed the web content in plain language and we structured it in layers and sections, combining text with other formats (images, static and interactive tables, symbols, videos in some cases, and others), with the aim of making it more user-friendly and understandable. We included the following sections in each evaluation [17]:

**Table 1. Factors of the certainty of the evidence.**

| Study design | Certainty of the evidence grade (initial) | Factors that can reduce the certainty of the evidence | Factors that can increase the certainty of the evidence | Certainty of the evidence grade (final) |
|---|---|---|---|---|
| Randomized trial | High | Risk of bias (study limitations) | Large magnitude of effect | High |
| | | Inconsistency of results (or heterogeneity) | Dose-response gradient | Moderate |
| | | Indirectness of evidence (PICO and applicability) | All plausible confounding and bias, which would reduce a demonstrated effect or would suggest a spurious effect if no effect was observed | |
| Observational study | Low | Imprecision (number of events and confidence intervals) | | Low |
| | | Publication bias | | Very low |

**Table 2. The certainty of the evidence and the veracity of the claims evaluated.**

| Claim | Certainty* | Statement# | Veracity |
|---|---|---|---|
| Desirable claim (claim with a beneficial effect) | High | Reduces/improves the risk | True |
| | | | False |
| Undesirable claim (claim with a harmful effect) | Moderate | Probably reduces/improves the risk | Probably true |
| | | | Probably false |
| | Low | May reduce/improve the risk | Possibly true |
| | | | Possibly false |
| | Very low | Uncertain whether improves/reduces the risk | Uncertain |

***High certainty** means that the authors have a lot of confidence that the true effect is similar to the estimated effect; **moderate certainty** means that the authors believe that the true effect is probably close to the estimated effect; **low certainty** means that the true effect might be markedly different from the estimated effect, and **very low certainty** means that the true effect is probably markedly different from the estimated effect. GRADE certainty ratings taken from BMJ Best Practice. What is GRADE? September 2019. Available: https://bestpractice.bmj.com/info/us/toolkit/learn-ebm/what-is-grade/

#The phrase of the statement should be elaborated taking into consideration the magnitude of the desirable and undesirable effects (see Table 3 and S3 Table).

- Headline: it summarizes the result of the evaluation in a sentence or, in some cases, it poses the question of the evaluation in plain language.

- Introduction and contextualization: we provided a brief introduction and contextualization about the claim being analysed.

- Conclusion: we stated the result of the evaluation with its corresponding symbol.

- Summary: we summarized the evaluation in plain language making the full scientific report available in PDF format. For some claims, we also produced a short video explaining the evaluation [18].

- What does the evidence say? (*Qué dice la ciencia*): this section provides a reasoned explanation of the certainty of the research results. When the result of the evaluation was not uncertain, an interactive SoF table was also provided. SoF tables are user-friendly formats to communicate research findings to the public and other stakeholders [19].

- To know more (*Para saber más*): this section includes relevant resources and links related to the evaluated claim.

The Nutrimedia website (https://www.upf.edu/web/nutrimedia) is responsive and multi-layered. It is hosted by Pompeu Fabra University, which provided the technical support for its development and maintenance.

## Media impact and website traffic analysis

We used two main methods to assess the media impact of Nutrimedia: the Factiva database, which includes newspapers, magazines and news agencies from all over the world; and the media monitoring service Acceso 360. The estimated economic value of each piece of news, provided by Acceso 360, was calculated based on the cost of the advertising space it occupies in a particular newspaper. We used Google Analytics to analyse website traffic.

## Results

Nutrimedia was launched in Spanish in November 20, 2017. During the first year and a half, we published 30 evaluations, of which 12 (40%) were questions from the public, 9 (30%) were

common beliefs, 7 (23%) were newspaper claims, and 2 (7%) were advertising claims. Most evaluations were related to raw or processed foods (70%; 21/30), followed by diets (20%; 6/30) and supplements (10%; 3/30); and the majority of them were based on SR (80%; 24/30). Nearly half of the claims were classified as "uncertain" (47%; 14/30), followed by "possibly true" (13%; 4/30), probably true (13%; 4/30), "probably false" (10%; 3/30), "possibly false" (7%; 2/30), false (7%; 2/30) and "true" (3%; 1/30) (Table 3). An example of the evaluation of the veracity of a claim is presented in the S4 Table.

Nutrimedia has other contents such as "Eating with science" (*Comer con ciencia*), "To know more" (*Para saber más*), and "About Nutrimedia" (*Sobre Nutrimedia*). The "To know more" section provides short videos to promote critical thinking and to facilitate the understanding of some methodological concepts (e.g., what is the GRADE approach and how certainty of the evidence was evaluated; guidelines for the public to interpret information on nutrition; guidelines for journalists to report on nutrition; relevant links in the field of nutrition, and a glossary). In the section "About Nutrimedia", users can find out who we are, the scientific methodology that we applied to evaluate claims on nutrition, the press releases and the media coverage.

In the section "Ask Nutrimedia", available from November 20, 2017, to May 5, 2019, the public assessed the interest of 10 questions in an online survey. Of the 12 questions from the public that were evaluated, 6 were the most voted of the 10 closed-ended questions and 6 were posed by users in the survey space for comments and suggestions. Some other questions about nutrition of general interest, most of them posed by the public, were answered by experts in articles and interviews (provided as podcasts) in the "Eating with science" section. The number of online survey respondents was 333 (55.2%, 182/333, general public; and 45.8%, 151/333, health professionals). Fifty-eight respondents (17.4%, 58/333) provided positive (51.7%, 30/58) (e.g., "An excellent idea to combat all the misinformation"; "Thank you for helping me get back to believing in science") and neutral (5.2%, 3/58%) comments or suggestions (43.1%, 25/58) (e.g., "It would be interesting to receive web updates by email"; "It would be tremendously helpful to give healthy and tasty recipes").

The number of website users was 47,265 (46,052 new users) since the website was launched (November 20, 2017) and for the first 18 months (until May 20, 2019). The number of page views in this period was 181,360. During this period, Nutrimedia was cited by 94 newspapers, news agencies and newswires included in Factiva Dow Jones database; of these 94 citations, 55 were in Spanish, 33 in Catalan, 4 in English and 2 in Portuguese. According to the data provided by Acceso 360, Nutrimedia was cited 78 times by newspapers and 386 times by websites from Spain and 14 other countries, mostly in Latin America. In addition, the Nutrimedia project and its evaluations were broadcasted on several Spanish radio and television channels. In that period, the estimated economic value of the 78 pieces of news published in printed editions of newspapers was €152,507.55.

Nutrimedia has also been cited by prominent Spanish dieticians and nutritionists. For example, Nutrimedia is referenced in 5 of its 12 recommendations for the general public in the 2018 food-based dietary guideline "Small changes to eating better" by the Public Health Agency of Catalonia [20].

Finally, Nutrimedia was well placed in nutrition searches in Spanish with Google, whose algorithm rates the pages according to the criteria of expertise, authority and trustworthiness [21]; e.g., in a Google search with the terms "meat cancer" (search performed on 2019 September 30, using a logged-out Chrome browser cleared of cookies and previous search history), Nutrimedia was ranked as the third listed information source in a list headed by the World Health Organization.

**Table 3. Type and veracity of the evaluated claims in https://www.upf.edu/web/nutrimedia until May 20, 2019.**

| Claim* | Type | Intervention | Critical outcome(s)# | Certainty of evidence | Statement(s) | Veracity of the claim |
|---|---|---|---|---|---|---|
| Moderate alcohol consumption is beneficial to health | Common beliefs | Alcohol | Breast cancer | High | Alcohol consumption (in any amount) increases breast cancer risk | False |
| To stay healthy is better to eat more than five daily servings of fruits and vegetables | Newspaper claims | Five servings of vegetables and fruits daily | All-cause mortality | Moderate | Habitual consumption of vegetables and fruits probably reduces the risk of all-cause mortality | Probably true |
| Palm oil is more harmful to health than other similar fats | Questions from the public | Palm oil vs similar fats (vegetable oils/ partially hydrogenated oils/ animal oils) | Cardiovascular disease (lipids levels) | Very low | We are uncertain whether palm oil increases the risk of cardiovascular disease in comparison of other similar fats | Uncertain |
| Danacol lowers high cholesterol up to 10% | Advertising claims | Danacol | LDL cholesterol | High | Danacol reduces the level of LDL cholesterol | True |
| Antioxidant supplements prevent diseases | Common beliefs | Antioxidant supplements | Mortality/ cardiovascular diseases / cancer | Low | Antioxidant supplements consumption may reduce the risk of all-cause mortality, cardiovascular disease and cancer | Probably false |
| Added sugar to food is harmful to health | Newspaper claims | Added sugar | Coronary diseases | Moderate | Added sugar to food probably increases the risk of coronary disease | Probably true |
| Gluten-free diet is beneficial for the health of healthy adults | Questions from the public | Gluten-free diet | Coronary diseases | Low | Gluten-free diet probably slightly reduces the risk of coronary disease | Probably false |
| Chocolate consumption prevents cardiovascular disease | Newspaper claims | Chocolate | Mortality from cardiovascular diseases/ cardiovascular disease | Very low | We are uncertain whether chocolate consumption reduces the risk of cardiovascular disease and cardiovascular mortality | Uncertain |
| Soy products are effective in treating the symptoms associated with menopause | Common beliefs | Soy products | Menopause symptoms (vasomotor and vaginal symptoms)/ cognitive function | Very low | We are uncertain whether soy products and soy supplements consumption reduces the risk of menopause symptoms | Uncertain |
| Coffee consumption is harmful to health | Newspaper claims | Coffee | Mortality/ Mortality from cardiovascular disease/ cardiovascular disease/ cancer | Very low | We are uncertain whether coffee consumption increases the risk of all-cause mortality, cardiovascular mortality, cardiovascular disease and cancer | Uncertain |
| White bread favors overweight more than whole wheat bread | Questions from the public | White bread vs whole bread | Body weight / abdominal circumference | Low | White bread may make little or no difference to weight and abdominal circumference/ Whole wheat bread may make no difference to weight and abdominal circumference | Possibly true |
| Energy drinks consumption counteracts the cognitive effects of alcohol consumption | Common beliefs | Energy drinks and alcohol/ Energy drinks/ Alcohol | Injuries/ cognitive function/ and behavioral disorders | Moderate | Alcohol mixed with energy drinks consumption probably no reduces the negative cognitive effects of alcohol consumption | Probably false |
| Light food products consumption reduces weight | Common beliefs | Low-fat foods/ low-calorie foods (light food products) | Body weight | Very low | We are uncertain whether low-fat foods reduces body weight | Uncertain |
| Omega-3 fatty acid supplements help prevent dementias | Questions from the public | Omega-3 supplements | Dementia | Low | Omega-3 supplements may make little or no difference to develop dementia | Probably false |
| It's just as healthy to drink a fruit juice than a whole fruit intake | Common beliefs | Fruit juices (100% fruit)/ Fruit juices with added sugar | Body weight, diabetes and cardiovascular risk | Very low | We are uncertain whether fruit juices increases or reduces body weight and the risk of diabetes and cardiovascular | Uncertain |

*(Continued)*

**Table 3.** (Continued)

| Claim* | Type | Intervention | Critical outcome(s)# | Certainty of evidence | Statement(s) | Veracity of the claim |
|---|---|---|---|---|---|---|
| Artificial sweeteners are harmful to health | Common beliefs | Artificial sweeteners | Diabetes/ obesity/ satiety and appetite | Moderate | Artificial sweeteners consumption slightly improves the metabolic control in diabetics patients, supervised by healthcare professionals/ Artificial sweeteners consumption (instead sugar) slightly reduces (or maintain) body weight in structured weight-loss programs supervised by healthcare professionals/ Artifical sweeteners consumption probably make little or no difference to hormonal regulators of satiety and appetite | Probably false |
| Vegan diet is beneficial to health | Questions from the public | Vegan diet/ Vegetarian diet | All-cause mortality/ cancer | Very low | We are uncertain whether vegan diet reduces the risk of all-cause mortality and cancer | Uncertain |
| Lactose free milk is easier to digest | Advertising claims | Lactose free milk | Gastrointestinal symptoms and diseases | No studies | No studies were found between lactose free milk consumption and gastrointestinal symptoms and diseases | Uncertain |
| Alkaline diet prevents cancer | Questions from the public | Alkaline diet | Cancer | Very low | We are uncertain whether alkaline diet reduces the risk of cancer | Uncertain |
| Meat is carcinogenic | Questions from the public | Meat/ red meat/ processed meat | Colorectal cancer | Low | Habitual red meat consumption may increase the risk of colorectal cancer/ Habitual processed meat probably increases the risk of colorectal cancer | Possibly true |
| Intermittent fasting is beneficial to health | Questions from the public | Intermittent fasting | Coronary artery disease/ diabetes | Very low | We are uncertain whether intermittent fasting reduces the risk of coronary artery disease and diabetes | Uncertain |
| Breastfeeding prevents obesity | Questions from the public | Breastfeeding | Obesity and overweight | Low | Breastfeeding may reduce the risk to develop obesity and overweight | Possibly true |
| Habitual nuts intake reduces cardiovascular risk | Newspaper claims | Nuts | All-cause mortality/ cardiovascular disease | Moderate | Habitual nuts intake probably reduces the risk of all-cause mortality and cardiovascular disease | Probably true |
| Habitual garlic intake helps prevent cancer | Questions from the public | Garlic | Cancer | Very low | We are uncertain whether habitual garlic intake reduces the risk to develop cancer | Uncertain |
| Mediterranean diet reduces the risk of depression | Newspaper claims | Mediterranean diet/ other healthy diets | Depression | Very low | We are uncertain whether mediterranean diet and other healthy diets reduce the risk of depression | Uncertain |
| Vitamin D supplements reduce the risk of fracture | Common beliefs | Vitamin D suplements | Fracture | High | Vitamin D no reduces the risk of fracture | False |
| Dairy intake helps prevent cardiovascular disease | Newspaper claims | Dairy | Cardiovascular disease | Low | Dairy intake (>two servings/day vs zero) may reduce the risk of cardiovascular disease | Possibly true |
| To lose weight, it is better to consume olive oil than other oils | Questions from the public | Olive oil | Body weight | Moderate | Olive oil consumption (instead other oils or fats) probably reduces body weight | Probably true |
| Daily egg consumption increases the risk of cardiovascular diseases | Common beliefs | Egg | Cardiovascular disease | Low | We are uncertain whether habitual egg consumption (one serving/day) increases the risk of cardiovascular disease | Uncertain |

(*Continued*)

**Table 3.** (Continued)

| Claim[*] | Type | Intervention | Critical outcome(s)[#] | Certainty of evidence | Statement(s) | Veracity of the claim |
|---|---|---|---|---|---|---|
| Ecological foods intake is beneficial for health | Questions from the public | Ecological foods | Cancer | Very low | We are uncertain whether ecological foods consumption (instead conventional foods) reduces the risk of cancer | Uncertain |

[*]Chronologically ordered from the oldest to the newest published on the web.

[#]Only those outcomes that we deemed as critical.

## Discussion

Nutrimedia is the first web-based resource for the general public that evaluates the veracity of media claims about nutrition based on the certainty of evidence, and to communicate the results in plain language and friendly presentation formats. We have evaluated 30 claims and classified nearly half of them as uncertain based on GRADE. During the first year and a half, Nutrimedia had a considerable impact on the media (about one impact per day).

There are some other online resources with different purposes and target populations. "Practice-based evidence in nutrition" provides the latest evidence in practice-based nutrition questions applying the GRADE approach, but this evidence-based decision support system is intended for dieticians/nutritionists and students [21]. "Behind the Headlines", from the UK National Health Service, analyses critically media claims related to nutrition, but does not assess the certainty of the evidence [22]. Cochrane Nutrition provides nutrition-related Cochrane systematic reviews, but there is often a communication gap between these reviews and the public [23]. To avoid this, Cochrane centers implement alternative ways to make evidence accessible in plain language summaries [24] and blogshots (infographics that summarize the evidence) in English [25], Spanish [26] and other languages. By overcoming these limitations, Nutrimedia is an innovative resource for disseminating quality nutrition information, and making it accessible to the public. Therefore, we believe that Nutrimedia is a pioneering initiative that promotes critical thinking and can have an impact on the food and nutrition literacy of the general public [27].

Nutrimedia has several strengths. The website is user-friendly because no registration is required, the evaluations are accessible within a few clicks and its content is multi-layered and multiformat. For each evaluation, we have applied a rigorous and explicit methodology using GRADE. GRADE approach represents a systematic, explicit and transparent methodological framework for grading the certainty of evidence and it has already been endorsed or used by over 100 organizations, the World Health Organization and the Cochrane Collaboration [28–29] among others. Finally, our project team has extensive knowledge and experience in the fields of nutrition, evidence-based medicine, methods, communication and journalism.

Nutrimedia also has some limitations. Firstly, it is only available in Spanish. Secondly, we used pragmatic search strategies favoring precision over sensitivity [30]. Thirdly, this project has no system yet for monitoring and continuous updating of the evaluations. Finally, analysis of its impact, usefulness, accessibility and understandability is still limited.

### Implications for practice and research

General public, journalists and communicators can use Nutrimedia to stay informed and making informed decisions about nutrition. Researchers interested in evaluating topics about

nutrition can use our approach. More research is needed about the impact, usefulness, accessibility and understandability of Nutrimedia.

## Supporting information

**S1 Table. Resources and search strategies.**
(DOCX)

**S2 Table. Key reporting aspects of the evidence.**
(DOCX)

**S3 Table. Standardised statements about effect according to the GRADE approach.**
(DOCX)

**S4 Table. Text box.** An example of a scientific evaluation of a nutrition claim.
(DOCX)

## Acknowledgments

We would like to thank Darío Lopez, Andrea Juliana Sanabria, Mónica Ballesteros, Carolina Requeijo Lorenzo, Karla Salas Gama, Paulina Fuentes, Laura Martínez García, and Alba Irigoyen for their contribution in this project. We would also like to thank to Victoria Leo for her assistance with the English text.

## Author Contributions

**Conceptualization:** Montserrat Rabassa, Pablo Alonso-Coello, Gonzalo Casino.

**Data curation:** Montserrat Rabassa, Pablo Alonso-Coello, Gonzalo Casino.

**Formal analysis:** Montserrat Rabassa, Pablo Alonso-Coello, Gonzalo Casino.

**Funding acquisition:** Gonzalo Casino.

**Investigation:** Montserrat Rabassa, Pablo Alonso-Coello, Gonzalo Casino.

**Methodology:** Montserrat Rabassa, Pablo Alonso-Coello, Gonzalo Casino.

**Project administration:** Gonzalo Casino.

**Resources:** Pablo Alonso-Coello, Gonzalo Casino.

**Supervision:** Pablo Alonso-Coello, Gonzalo Casino.

**Validation:** Gonzalo Casino.

**Writing – original draft:** Montserrat Rabassa.

**Writing – review & editing:** Pablo Alonso-Coello, Gonzalo Casino.

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
