## [Decision Letter · Decision Letter 0]

3 Feb 2020

PONE-D-19-33111

Nutrimedia: A novel Spanish web-based resource for the general public that evaluates the credibility of nutritional messages using the GRADE approach

PLOS ONE

Dear Dr Casino,

Thank you for submitting your manuscript to PLOS ONE. After careful consideration, we feel that it has merit but does not fully meet PLOS ONE’s publication criteria as it currently stands. Therefore, we invite you to submit a revised version of the manuscript that addresses the points raised during the review process.

We would appreciate receiving your revised manuscript by Mar 07 2020 11:59PM. To enhance the reproducibility of your results, we recommend that if applicable you deposit your laboratory protocols in protocols.io, where a protocol can be assigned its own identifier (DOI) such that it can be cited independently in the future. For instructions see: http://journals.plos.org/plosone/s/submission-guidelines#loc-laboratory-protocols

We look forward to receiving your revised manuscript.

Kind regards,

David Meyre

Academic Editor

PLOS ONE

3. Thank you for stating the following in the Acknowledgments Section of your manuscript: "Nutrimedia has been partially supported by the Spanish Foundation for Science and Technology (FECYT) grants from the Spanish Ministry of Science, Innovation and Universities (FCT-16-11294 and FCT-17-12460). The funders had no role in study design, data collection and analysis, decision to publish, or preparation of the manuscript."

Please remove any funding-related text from the manuscript and let us know how you would like to update your Funding Statement. Currently, your Funding Statement reads as follows: "The funders had no role in study design, data collection and analysis, decision to publish, or preparation of the manuscript."

4. Please include a copy of Table 4 which you refer to in your text on page 10.

Reviewers' comments:

Reviewer's Responses to Questions

**Comments to the Author**

1. Is the manuscript technically sound, and do the data support the conclusions?

Reviewer #1: Yes

Reviewer #2: Partly

2. Has the statistical analysis been performed appropriately and rigorously? 

Reviewer #1: N/A

Reviewer #2: N/A

3. Have the authors made all data underlying the findings in their manuscript fully available?

Reviewer #1: No

Reviewer #2: Yes

4. Is the manuscript presented in an intelligible fashion and written in standard English?

Reviewer #1: Yes

Reviewer #2: Yes

5. Review Comments to the Author

Reviewer #1: This paper describes the development, characteristics, main contents and media impact of Nutrimedia, a web-based resource for the public that evaluates the certainty of nutrition messages using the GRADE framework. This is a noble effort, as though GRADE is known to those of us who work in guidelines development, and often included in systematic reviews that make headlines, rarely is the GRADE associated with the findings reported.

I have the following suggestions that I hope will help improve the manuscript:

1. Line 91: How do you determine which myths to test? And how did you ascertain that the "myth" was indeed myth prior to assessing the evidence? The word "myth" carries a judgement with it. It seems to me that you might call this a "statement" or "commonly held belief", which you then classify as myth/fact after your evidence review, based on your judgement on the 7 categories. If you have already made the judgement that a commonly held belief is a "myth" before evaluating the evidence, then why evaluate it?

2. Line 93: What is meant by "newspaper claims"? Are you referring to newspaper articles that report the findings of a nutrition study?

3. Lines 213-216: Please detail your assumptions and methodology for assessing the economic impact and potential audience. I do see the notes in parentheses, but would like to see you reference the source of these numbers and how you estimated the final figures. How much uncertainty is there in these estimates?!

4. Lines 217-218: I don't follow how 58 of 333 is 56.9%. The values in parentheses that follow "(56.9%, 33/58; 90.1 % positive and 9.9% neutral comments) or suggestions (43.1%, 25/58)" ... are difficult to understand. For example, what is the denominator?

5. Lines 228-229: Are you able to provide some evidence for the claim that "During this period, this project and its results had considerable impact on the media."

6. Lines 107-108: It may be useful to provide a definition of a "pragmatic search", as readers may not be able to see the difference between this and a conventional systematic search. This is important, because you note this as a limitation in discussion (lines 262-263).

7. Table 3. The statement "Alcohol increases with any amount of alcohol consumption breast cancer risk" is unclear to me. Could you rephrase?

8. Does your system provide a way to assess whether the studies supporting your credibility rating were funded by an industry with a "stake" in the claim? For example (and not making any accusation of impropriety)... your statement on Danacol's advertising claim was considered highly credible, based on high quality evidence (and I know that EFSA has given it the green light). But in the spirit of your initiative to evaluate credibility of advertising, would it be useful to share with your audience if Danone funded any of the trials? How do you deal with this concern in your system?

Reviewer #2: What are the main claims of the paper and how significant are they for the discipline?

- The Nutrimedia research project was an effort to improve public messaging about nutrition, which is thought to contribute to poor eating habits leading to increased morbidity and mortality. The authors used the GRADE approach to evaluate 30 messages from media and advertising, or asked by the public, about nutrition. Nutrition is a very popular topic for the general public, and inaccurate and confusing health information is an issue many stakeholders (e.g. consumers, healthcare providers, policy makers) are concerned about.

Are the claims properly placed in the context of the previous literature? Have the authors treated the literature fairly?

- The authors noted several resources that also evaluate nutrition claims, which was useful. However, I think there could have been more contextualization of the project. For example, it was unclear whether having higher-quality information would have led to appreciable changes in people’s behaviours/health - it may be helpful to find research supporting that access to/knowledge of more accurate healthcare information leads to better health outcomes.

Do the data and analyses fully support the claims? If not, what other evidence is required?

- There are several supportive references in the introduction about the importance of media in affecting people's knowledge and awareness of nutrition. However, I felt that the authors were too focused on the lack of a comprehensive website about nutrition as being the most important reason the Nutrimedia project was created. It would be helpful to have more supporting literature identifying a use case for the project.

PLOS ONE encourages authors to publish detailed protocols and algorithms as supporting information online. Do any particular methods used in the manuscript warrant such treatment? If a protocol is already provided, for example for a randomized controlled trial, are there any important deviations from it? If so, have the authors explained adequately why the deviations occurred?

- The authors publish an outline of their methods, and a full protocol is available in Spanish to those who request it.

If the paper is considered unsuitable for publication in its present form, does the study itself show sufficient potential that the authors should be encouraged to resubmit a revised version?

- I think the paper is publishable already, and is about a topic of interest to a wide readership. However, it could be improved with more detail about the methodology and more context about the potential impact of the project.

Are original data deposited in appropriate repositories and accession/version numbers provided for genes, proteins, mutants, diseases, etc.?

- Not applicable.

Are details of the methodology sufficient to allow the experiments to be reproduced?

Is the manuscript well organized and written clearly enough to be accessible to non-specialists?

- There needs to be more description about the methodology. See below for suggestions in major comments.

Major comments:

1. There should be more detail to describe the novel methodological approach to identifying topics and evaluating evidence. Specifically:

- What was the criteria for choosing the most important messages (out of the 30)?

- Why were only 10 messages chosen to ask the public about relative importance?

- What’s the distinction between newspaper claims and advertising claims?

- How were questions from the public gathered (e.g. who was asked, time frame)? How many people responded?

- What did the grey literature search include, e.g. which governmental institutions and scientific societies were reviewed?

- You mention that you “generally avoided surrogate outcomes” but two of your questions were regarding surrogate outcomes (i.e. palm oil, dunacol) – can you clarify why this was done (e.g. public importance)?

- What kind of evidence was ultimately used to inform the certainty of each message?

- Did the SRs and CPGs you chose to inform the questions have to use GRADE to be eligible to inform the message?

2. In the limitations you mention “….these [other resources] are not brought together in a single, friendly online resource. By overcoming these limitations, to the best of our knowledge, Nutrimedia is an effective strategy for promoting scientific knowledge and awareness about nutritional messages that reaches the public through media and social networks.” I feel that there is too strong of a conclusion here about Nutrimedia's effectiveness. While Nutrimedia is a novel and interesting approach to improving health information accessible to the public, I was wondering whether lack of knowledge is the major gap (as opposed to not being able to afford more nutritious food, or other structural disadvantages unrelated to nutrition) and if Nutrimedia was the best way to address it. For future research, it would be informative to have qualitative/user-testing data on whether the novel presentation format is more accessible, understandable, and useful. I think this should be mentioned in the limitations.

Minor comments:

1. There are several mentions of Google searches, for example “For example, a recent Google search showed over half a million results related to the terms “nutrition advice”.” in the introduction, and “However, Nutrimedia appears as the third listed information source in a Google search with the terms “meat cancer” (search performed on September 30, 2019), after WHO and “mejorsincancer.org” (a scientific website related to cancer prevention Bellvitge Biomedical Research Institute (IDIBELL) - Catalan Institute of Oncology (ICO).” in the discussion. Perhaps this is because I am not familiar with research on media use, but it does not seem that strongly supportive.

2. This may be more of an editorial issue that will be changed at publication, but the number of links in the results about the Nutrimedia website sections was distracting.

3. The phrase “Online resources or websites (e.g., Google searches, YouTube) were the most popular source of nutrition information among adults (11-12).” should be clarified with the year(s), given that people’s information seeking behaviour likely changed in recent years.

4. You used the phrase “friendly” to describe the format/presentation, but the first time this was explained was the strengths section, and it was because users did not need to register and the evaluation was accessible with few clicks. Of course this is important, but it would be helpful if this was detailed more in the methods section, and any other relevant information to support why you labeled the presentation formats as friendly.

5. I did not understand the economic impact statement.

6. The “333 online participants” section was unclear? Was this a summary of all public comments about the messages?

6. PLOS authors have the option to publish the peer review history of their article (what does this mean?). If published, this will include your full peer review and any attached files.

Reviewer #1: No

Reviewer #2: Yes: Lyubov Lytvyn

---

## [Author Response · Author response to Decision Letter 0]

6 Apr 2020

Reviewer #1: 

This paper describes the development, characteristics, main contents and media impact of Nutrimedia, a web-based resource for the public that evaluates the certainty of nutrition messages using the GRADE framework. This is a noble effort, as though GRADE is known to those of us who work in guidelines development, and often included in systematic reviews that make headlines, rarely is the GRADE associated with the findings reported.

I have the following suggestions that I hope will help improve the manuscript:

1. Line 91: How do you determine which myths to test? And how did you ascertain that the "myth" was indeed myth prior to assessing the evidence? The word "myth" carries a judgement with it. It seems to me that you might call this a "statement" or "commonly held belief", which you then classify as myth/fact after your evidence review, based on your judgement on the 7 categories. If you have already made the judgement that a commonly held belief is a "myth" before evaluating the evidence, then why evaluate it?

Response: We thank the reviewer for this observation. To avoid confusion, we have replaced the word “myth” with “common beliefs”. We have also clarified this point in the text (lines 107-108).

2. Line 93: What is meant by "newspaper claims"? Are you referring to newspaper articles that report the findings of a nutrition study?

Response: As the reviewer suggests, we have added and clarified this point in the text as follows: “Claims in pieces of news published in newspapers that report about nutrition or nutrition research (newspaper claims).”

3. Lines 213-216: Please detail your assumptions and methodology for assessing the economic impact and potential audience. I do see the notes in parentheses, but would like to see you reference the source of these numbers and how you estimated the final figures. How much uncertainty is there in these estimates?!

Response: As suggested by the reviewer, we have now included this point in the text (lines 210-216).

4. Lines 217-218: I don't follow how 58 of 333 is 56.9%. The values in parentheses that follow "(56.9%, 33/58; 90.1 % positive and 9.9% neutral comments) or suggestions (43.1%, 25/58)" ... are difficult to understand. For example, what is the denominator?

Response: We have revised the values and rewritten the sentence as follows: “The number of online survey respondents was 333 (55.2%, 182/333, general public; and 45.8%, 151/333, health professionals). Fifty-eight respondents (17.4%, 58/333) provided positive (51.7%, 30/58) (e.g., “An excellent idea to combat all the misinformation”; “Thank you for helping me get back to believing in science”) and neutral (5.2%, 3/58%) comments or suggestions (43.1%, 25/58) (e.g., “It would be interesting to receive web updates by email”; It would be tremendously helpful to give healthy and tasty recipes”).”

5. Lines 228-229: Are you able to provide some evidence for the claim that "During this period, this project and its results had considerable impact on the media."

Response: As suggested by the reviewer, we have now clarified this information in the text (lines 273-304; 312-313).

6. Lines 107-108: It may be useful to provide a definition of a "pragmatic search", as readers may not be able to see the difference between this and a conventional systematic search. This is important, because you note this as a limitation in discussion (lines 262-263).

Response: As the reviewer suggests, we have now added the definition of a “pragmatic search” as follows: “we used pragmatic search strategies favoring precision over sensitivity (30).”

7. Table 3. The statement "Alcohol increases with any amount of alcohol consumption breast cancer risk" is unclear to me. Could you rephrase?

Response: As the reviewer suggests, we have now rephrased this statement as follows: “Alcohol consumption (in any amount) increases breast cancer risk”. 

8. Does your system provide a way to assess whether the studies supporting your credibility rating were funded by an industry with a "stake" in the claim? For example (and not making any accusation of impropriety)... your statement on Danacol's advertising claim was considered highly credible, based on high quality evidence (and I know that EFSA has given it the green light). But in the spirit of your initiative to evaluate credibility of advertising, would it be useful to share with your audience if Danone funded any of the trials? How do you deal with this concern in your system?

Response: The GRADE system provides a way to assess publication bias, a more common factor when most of the published studies (trials and/or observational) are funded by industry. On the Danacol’s advertising claim, we have not observed evidence of publication bias through statistical and visual methods. An example of this was provided by a meta-analysis of 59 eligible randomized clinical trials (AbuMweis 2008). 

- AbuMweis et al. Plant sterols/stanols as cholesterol lowering agents: A meta-analysis of randomized controlled trials. Food Nutr Res. 2008; 52: 10.3402/fnr.v52i0.1811.

\f

Reviewer #2: 

What are the main claims of the paper and how significant are they for the discipline?

- The Nutrimedia research project was an effort to improve public messaging about nutrition, which is thought to contribute to poor eating habits leading to increased morbidity and mortality. The authors used the GRADE approach to evaluate 30 messages from media and advertising, or asked by the public, about nutrition. Nutrition is a very popular topic for the general public, and inaccurate and confusing health information is an issue many stakeholders (e.g. consumers, healthcare providers, policy makers) are concerned about.

Are the claims properly placed in the context of the previous literature? Have the authors treated the literature fairly?

- The authors noted several resources that also evaluate nutrition claims, which was useful. However, I think there could have been more contextualization of the project. For example, it was unclear whether having higher-quality information would have led to appreciable changes in people’s behaviours/health - it may be helpful to find research supporting that access to/knowledge of more accurate healthcare information leads to better health outcomes.

Do the data and analyses fully support the claims? If not, what other evidence is required?

- There are several supportive references in the introduction about the importance of media in affecting people's knowledge and awareness of nutrition. However, I felt that the authors were too focused on the lack of a comprehensive website about nutrition as being the most important reason the Nutrimedia project was created. It would be helpful to have more supporting literature identifying a use case for the project.

PLOS ONE encourages authors to publish detailed protocols and algorithms as supporting information online. Do any particular methods used in the manuscript warrant such treatment? If a protocol is already provided, for example for a randomized controlled trial, are there any important deviations from it? If so, have the authors explained adequately why the deviations occurred?

- The authors publish an outline of their methods, and a full protocol is available in Spanish to those who request it.

If the paper is considered unsuitable for publication in its present form, does the study itself show sufficient potential that the authors should be encouraged to resubmit a revised version?

- I think the paper is publishable already, and is about a topic of interest to a wide readership. However, it could be improved with more detail about the methodology and more context about the potential impact of the project.

Are original data deposited in appropriate repositories and accession/version numbers provided for genes, proteins, mutants, diseases, etc.?

- Not applicable.

Are details of the methodology sufficient to allow the experiments to be reproduced?

Is the manuscript well organized and written clearly enough to be accessible to non-specialists?

- There needs to be more description about the methodology. See below for suggestions in major comments.

Major comments:

1. There should be more detail to describe the novel methodological approach to identifying topics and evaluating evidence. Specifically:

- What was the criteria for choosing the most important messages (out of the 30)?

- Why were only 10 messages chosen to ask the public about relative importance?

- What’s the distinction between newspaper claims and advertising claims?

Response: We thank the reviewer for these observations. We have added and clarified this information in the methods section (lines 100-128). 

- How were questions from the public gathered (e.g. who was asked, time frame)? How many people responded?

Response: We thank the reviewer for these observations. We have added and clarified this information in the text (lines 121-128; 240-248; 249-255). 

- What did the grey literature search include, e.g. which governmental institutions and scientific societies were reviewed?

Response: We have now specified it in the text as follows: “We also searched grey literature (e.g. Google Scholar), governmental and institutional sources (e.g. World Health Organization, U.S. Department of Agriculture, Spanish Agency for Food Safety and Nutrition) and scientific societies websites (e.g., Sociedad Española de Nutrición Comunitaria).”

- You mention that you “generally avoided surrogate outcomes” but two of your questions were regarding surrogate outcomes (i.e. palm oil, dunacol) – can you clarify why this was done (e.g. public importance)?

Response: In the case of Danacol claim “Danacol lowers high cholesterol up to 10%”, the key public-critical (or important) outcome for the PICO question was related directly with a surrogate outcome, named LDL cholesterol. Instead, in the case of palm oil “Palm oil is more harmful to health than other similar fats”, the key public-critical outcome of interest was cardiovascular disease. However, no studies were found that looked at cardiovascular disease and/or mortality. Therefore, the outcome was represented by surrogate outcomes (concretely by total, LDL and HDL cholesterol), which decreases the quality of evidence part on one side for indirectness. 

To avoid confusion, we have removed the sentence on surrogate outcomes.

- What kind of evidence was ultimately used to inform the certainty of each message?

Response: As mentioned in the previous response, the kind of evidence for each claim that we used to inform the certainty was related with the key public-critical outcome (this is specified in Table 3). 

- Did the SRs and CPGs you chose to inform the questions have to use GRADE to be eligible to inform the message?

Response: We thank the reviewer for this observation. In order to avoid confusion, we have replaced “quality” with “risk of bias” (line 150).

2. In the limitations you mention “….these [other resources] are not brought together in a single, friendly online resource. By overcoming these limitations, to the best of our knowledge, Nutrimedia is an effective strategy for promoting scientific knowledge and awareness about nutritional messages that reaches the public through media and social networks.” I feel that there is too strong of a conclusion here about Nutrimedia's effectiveness. While Nutrimedia is a novel and interesting approach to improving health information accessible to the public, I was wondering whether lack of knowledge is the major gap (as opposed to not being able to afford more nutritious food, or other structural disadvantages unrelated to nutrition) and if Nutrimedia was the best way to address it. For future research, it would be informative to have qualitative/user-testing data on whether the novel presentation format is more accessible, understandable, and useful. I think this should be mentioned in the limitations.

Response: We thank the reviewer for this important comment. We have properly re-edited the conclusion as follows: “Nutrimedia is an innovative resource for disseminating quality nutrition information, and making it accessible to the public.” 

Additionally, as suggested by the reviewer, we have now rewritten this information in the limitations as follows: “Analysis of its impact, usefulness, accessibility and understandability is still limited.”

Minor comments:

1. There are several mentions of Google searches, for example “For example, a recent Google search showed over half a million results related to the terms “nutrition advice”.” in the introduction, and “However, Nutrimedia appears as the third listed information source in a Google search with the terms “meat cancer” (search performed on September 30, 2019), after WHO and “mejorsincancer.org” (a scientific website related to cancer prevention Bellvitge Biomedical Research Institute (IDIBELL) - Catalan Institute of Oncology (ICO).” in the discussion. Perhaps this is because I am not familiar with research on media use, but it does not seem that strongly supportive.

Response: As the reviewer suggests, we have now properly supported this point in the text (lines 299-304).

2. This may be more of an editorial issue that will be changed at publication, but the number of links in the results about the Nutrimedia website sections was distracting.

Response: As the reviewer and editor suggest, we have now removed these links. 

3. The phrase “Online resources or websites (e.g., Google searches, YouTube) were the most popular source of nutrition information among adults (11-12).” should be clarified with the year(s), given that people’s information seeking behaviour likely changed in recent years.

Response: As suggested by the reviewer, we have clarified this information in the text (line 87). 

4. You used the phrase “friendly” to describe the format/presentation, but the first time this was explained was the strengths section, and it was because users did not need to register and the evaluation was accessible with few clicks. Of course this is important, but it would be helpful if this was detailed more in the methods section, and any other relevant information to support why you labeled the presentation formats as friendly.

Response: We have now described “friendly” in the methods section (lines 186-190; 202-204). Additionally, we have rewritten the sentence in the strengths section to clarify this information (lines 334-336).

5. I did not understand the economic impact statement.

Response: We have now added the information related with this point in the methods (lines 210-216).

6. The “333 online participants” section was unclear? Was this a summary of all public comments about the messages?

Response: We thank the reviewer for these observations. Regarding the first question, we have clarified this information in the results (lines 249-255). For the second question, we have now added some examples of public comments about the claims also in the results (lines 251-255).

---

## [Editor Report · Decision Letter 1]

15 Apr 2020

Nutrimedia: A novel web-based resource for the general public that evaluates the veracity of nutrition claims using the GRADE approach

PONE-D-19-33111R1

Dear Dr. Casino,

We are pleased to inform you that your manuscript has been judged scientifically suitable for publication and will be formally accepted for publication once it complies with all outstanding technical requirements.

With kind regards,

David Meyre

Academic Editor

PLOS ONE
---

## [Editor Report · Acceptance letter]

20 Apr 2020

PONE-D-19-33111R1 

Nutrimedia: A novel web-based resource for the general public that evaluates the veracity of nutrition claims using the GRADE approach 

Dear Dr. Casino:

I am pleased to inform you that your manuscript has been deemed suitable for publication in PLOS ONE. Congratulations! Your manuscript is now with our production department. 

With kind regards,

on behalf of

Dr David Meyre 

Academic Editor

PLOS ONE